# Single Slow-Paced Breathing Session at Six Cycles per Minute: Investigation of Dose-Response Relationship on Cardiac Vagal Activity

**DOI:** 10.3390/ijerph182312478

**Published:** 2021-11-26

**Authors:** Min You, Sylvain Laborde, Nina Zammit, Maša Iskra, Uirassu Borges, Fabrice Dosseville

**Affiliations:** 1EA 3918 CERREV, UFR Psychologie, Université de Caen Normandie, 14000 Caen, France; min.you@unicaen.fr; 2Department of Performance Psychology, Institute of Psychology, German Sport University Cologne, 50933 Cologne, Germany; ninazt27@gmail.com (N.Z.); masa.iskra@gmail.com (M.I.); u.borges@dshs-koeln.de (U.B.); 3EA 4260 CESAMS, UFR STAPS, Université de Caen Normandie, 14000 Caen, France; 4Department of Health & Social Psychology, Institute of Psychology, German Sport University Cologne, 50933 Cologne, Germany; 5UMR-S 1075 COMETE, INSERM, Université de Caen Normandie, 14000 Caen, France; fabrice.dosseville@unicaen.fr

**Keywords:** heart rate variability, parasympathetic nervous system, vagus nerve, diaphragmatic breathing, abdominal breathing

## Abstract

The practice of slow-paced breathing (SPB) has been linked to a range of positive outcomes, such as decreasing symptoms of depression, anxiety, and stress, as well as increasing well-being. Among the suggested mechanisms of action, SPB has been shown to increase cardiac vagal activity (CVA). The present study aimed to investigate whether there is a dose-response relationship modulating the effects of SPB on CVA. A total of 59 participants were involved in this study. In a within-subject design, participants attended the lab five times, and realized SPB at six cycles per minute with different durations (5, 10, 15, and 20 min), as well as a control condition without SPB. CVA was indexed via the root mean square of successive differences (RMSSD). During SPB, findings showed an increase in RMSSD in all conditions compared to the control condition. However, no differences were found in RMSSD among the different session durations, during SPB or during the resting measurement completed immediately after SPB. Noteworthily, session duration showed an influence on the spontaneous respiratory frequency in the resting measurement occurring immediately after SPB. Specifically, respiratory frequency appears to decrease with session duration, thus potentially contributing to additional relaxing effects.

## 1. Introduction

Slow-paced breathing (SPB), the voluntary slowing down of respiratory frequency (RF), is linked to a range of positive mental and physical health outcomes, such as decreased depression, anxiety, and stress symptoms [1,2,3]. Different parameters have been found to differentially influence the effectiveness of SPB, for example, the inhalation/exhalation ratio. A longer exhalation in relation to inhalation was found to trigger more psychophysiological benefits [4,5,6]. Another aspect which has not yet received researchers’ attention to date is the dose-response relationship. Specifically, whether the duration of SPB also influences its psychophysiological effects. Investigating the dose-response (also called exposure-response) relationship is a typical approach to explore the effectiveness of interventions and treatments [7]. Therefore, the aim of this paper is to address the dose-response relationship regarding the duration of a single SPB session.

While the spontaneous breathing frequency usually comprises between 12 and 20 cycles per minute [8,9], SPB aims to slow down breathing frequency to around six cycles per minute (cpm). Pacing breathing refers to controlling the duration of the inhalation and exhalation phases. The mechanisms underlying the effects of SPB are still debated. However, these are likely to involve the strengthening of the baroreflex [10,11], the increase in activity of pulmonary afferents [12], and the strengthening of the connectivity among brain networks involved in emotion regulation [13]. Overall, the stimulation of the vagus nerve, the main nerve of the parasympathetic nervous system [14], has been suggested as a common underlying mechanism for the effects of SPB [2,3].

The activity of the vagus nerve regulating cardiac functioning, referred to as cardiac vagal activity (CVA), has been shown to be involved in important self-regulation mechanisms of the organism [15,16]. The neurovisceral integration model [17,18], which is based on the central autonomic network [19], describes common brain structures which are involved in the regulation of emotion, cognition, and cardiac functioning. According to this model, CVA indexes the integrity of this regulation. CVA can be measured non-invasively via heart rate variability (HRV), the variation in the time intervals between successive heartbeats [20,21,22].

SPB has been shown to trigger increases in CVA, on an acute [23] and chronic basis [24]. A recent study has found no differences between SPB with and SPB without biofeedback regarding its effects on CVA [25]. Therefore, in this study, we focus particularly on SPB implemented without biofeedback, in which SPB is practiced without the live display of a biological signal [26], such as the RF, heart rate or HRV [11,27]. Concerning studies involving a single-session SPB without biofeedback and CVA, durations from 3 [28], 5 [4,5,29,30], 15 [23,31,32], and up to 30 min [33] have been used in previous research. Few studies adopted a design including CVA measurement during and after SPB, but existing findings indicate an increase in CVA during SPB with a return to baseline levels upon the cessation of SPB. This pattern could be observed with different durations of SPB, namely 5 min [30] or 15 min [23]. Due to the high interindividual differences in HRV [22,34], it appears worth investigating the dose-response relationship of SPB on CVA in a within-subject design, as the current study aims to do.

To sum up, the current study aimed to address the previous research gap concerning the dose-response relationship of a single SPB session realized without biofeedback on CVA. Specifically, we aim to systematically manipulate the duration of SPB, by applying durations of 5, 10, 15, and 20 min, in comparison to a control condition in which no SPB was performed. Based on recommendations for HRV investigation in psychophysiological experiments [22,34], a within-subject design was adopted. Through previous research, we hypothesized a condition effect, in which all SPB conditions would show a higher CVA during the intervention. No specific hypotheses were made regarding the dose-response relationship, due to the absence of a theoretical rationale or previous work on the topic. As a secondary outcome and manipulation check, RF will be measured.

## 2. Materials and Methods

### 2.1. Participants

In order to determine the sample size, we assumed a small-to-moderate effect size (*f* = 0.15) for the SPB dose-response effects on CVA. A G*Power [35], a priori power calculation with an effect size *f* = 0.15, Power (1-*β*) = 0.8, five measurements, and correlation among repeated measures = 0.50, provided an estimated sample size of 55. To anticipate for potential dropouts and technical issues, a sample size of *N* = 65 was recruited. Exclusion criteria were self-reported cardiovascular diseases, and other chronic diseases that might influence breathing or HRV patterns, such as asthma, diabetes, psychiatric, and neurological diseases [22]. Four participants dropped out of the experiment due to conflicting schedules, while two had to be excluded due to technical issues. The final sample size comprised *N* = 59 participants (*M*_age_ = 21.4, age range = 18–26 years old). The protocol of the study was approved by the Ethics committee of the local University (No. 105/2014).

### 2.2. Material and Measures

#### 2.2.1. Cardiac Vagal Activity

CVA was operationalized via HRV measurement, more specifically by the root mean square of successive differences (RMSSD), which has been found to index CVA [20,21,22] being relatively free of respiratory influences [36]. An electrocardiography (ECG) device (Faros 180°, Mega Electronics, Kuopio, Finland) was used to measure HRV, with a sampling rate of 500 Hz. We used two disposable ECG pre-gelled electrodes (Ambu L-00-S/25, Ambu GmbH, Bad Nauheim, Germany). The negative electrode was placed in the right infraclavicular fossa (just below the right clavicle), while the positive electrode was placed on the left side of the chest, below the pectoral muscle in the left anterior axillary line. RMSSD was extracted with Kubios (University of Eastern Finland, Kuopio, Finland). The full ECG recording was inspected visually, and artifacts were corrected manually [22]. In order to provide an overview of the different HRV parameters, following the recommendations by Laborde, Mosley et al. [22], we also extracted the heart rate and the standard deviation of the NN interval (SDNN) for the time-domain and low-frequency (LF: 0.04 to 0.15 Hz), high-frequency (HF: 0.15 to 0.40 Hz), and the LF/HF ratio were obtained for the frequency-domain (Fast Fourier Transform). Finally, the RF was extracted from the ECG signal, based on the ECG-derived respiration algorithm of Kubios [37].

#### 2.2.2. Slow-Paced Breathing

SPB was performed with a video showing a little ball moving up and down at the rate of 6 cpm. The video was recorded using the software EZ-Air Plus (Thought Technology Ltd., Montreal, QC, Canada), and was already used in previous research involving SPB without biofeedback [23,38]. Participants were instructed to inhale continuously through the nose while the ball was moving upwards, and exhale continuously with pursed lips when the ball was moving downwards. Inhaling via the nose (i.e., nasal breathing) makes the air warmer, cleaner, and more humid [39]. In addition, it may optimize various aspects of brain function [40,41,42]. Exhaling via the mouth offers less ventilatory resistance than the nasal channel [39]. Moreover, exhalation realized via pursed-lips offers greater control over the flow of air, enabling participants to match it precisely to the exhalation duration, which helps in reducing the respiratory rate [43]. Participants were then asked to put one hand on their chest and one hand on their stomach, and were given the following instructions [25,32]: “The hand on the chest should not move, only the hand on the belly should move: The belly should get bigger during the inhalation phase, and smaller during the exhalation phase”. This instruction reflects an optimal activation of the diaphragm. When the diaphragm contracts and goes down, it increases the volume of the thoracic cavity and creates an area of low pressure that causes air to flow into the lungs to equalize the pressure [39]. The video displayed a 1, 2, 3 or 4 × 5 min SPB exercise, with a 1-min break in between. Exhalation (6 s) was slightly longer than inhalation (4 s), given a longer exhalation phase was found to provoke a larger increase in CVA [4,5,6,44].

#### 2.2.3. TV Neutral Documentary

The control condition used an emotionally neutral TV documentary about world travel destinations, already used in previous research [45].

### 2.3. Procedure

Participants were recruited via flyers on the campus of the local university and via posts on social network groups linked to the local university. Participants had to attend five sessions in a randomized order. Randomization was performed using Research Randomizer [46], which is based on the “Math.random” method within the JavaScript programming language. The five sessions took place at the same time of the day, to account for daily variability in HRV [47], with at least 24 h intervals between the sessions. Prior to the testing sessions, participants were instructed not to drink or eat anything but water for 2 h before the experiment, as well as not to take part in any strenuous exercise or drink alcohol for the 24 h prior to testing [22]. Upon arrival at the laboratory, participants were asked to fill out an informed consent form and a demographic questionnaire regarding variables potentially influencing HRV [22]. At the beginning of the first condition involving SPB (thus in all conditions but the control condition), participants received a short video introduction on how to perform the technique correctly, which was checked by the experimenter. Additionally, a brief SPB technique check was undergone by the experimenter at the beginning of the other breathing sessions. In particular, the experimenter checked whether participants had their mouth closed while inhaling via the nose and were then exhaling via pursed-lips, whether they were closely following the timing of the breathing pacer, and whether their abdomen and not chest was moving during breathing. The ECG device was attached, and HRV was measured continuously throughout the experiment, in a sitting position, knees at 90°, hands on the thighs. All measures were conducted with eyes opened. An overview of the protocol can be seen in Figure 1.

### 2.4. Data Analysis

The ECG signal was imported into Kubios, and HRV variables were exported from the Kubios output. Data were checked for normality and outliers. Regarding outliers, 0.004% of the cases were found to be univariate outliers (>2 SD, z-scores higher than 2.58; none were found being >3 SD, with z-scores higher than 3.29). Running the analysis without them did not change the pattern of results, thus they were kept in the analysis. As the data were non-normally distributed, a log-transformation was applied, as it is usually recommended for the HRV research [22].

We conducted a series of repeated-measures ANOVA with Greenhouse-Geisser correction, with condition (CONTROL, 5, 10, 15, 20 min) and time (PRE, DURING, POST) as independent variables, with log RMSSD as a HRV-dependent variable indexing CVA, and then with RF as a manipulation check. Regarding the time point DURING, the given HRV measurements need to be of similar duration to enable a standardized comparison [20], we chose the last 5 min of each SPB session (i.e., 0 to 5 min for the 5 min condition; 5 to 10 min for the 10 min condition; 10 to 15 min for the 15 min condition; and 15 to 20 min for the 20 min condition). Post-hoc t-tests comparisons were run with Bonferroni correction to further explore the significant effects, reporting the effect size with Cohen’s d and the significance level.

## 3. Results

Descriptive statistics are presented in Table 1 for all study variables, and mean comparisons regarding the post-hoc tests are shown in Table 2 and Table 3.

### 3.1. Main Dependent Variable of Interest: Root Mean Square of Successive Differences (RMSSD)

For RMSSD, we found a significant main effect of condition *F*(2.575, 149.364) = 15.895, *p* < 0.001, partial *η*^2^ = 0.06; a significant main effect of time *F*(1.805, 104.697) = 91.251, *p* < 0.001, partial *η*^2^ = 0.31; and a significant time * condition interaction effect *F*(5.862, 340.002) = 32.611, *p* < 0.001, partial *η*^2^ = 0.08.

Regarding the main effect of condition, 10 follow-up post-hoc *t*-tests were conducted, adjusting alpha level with Bonferroni correction to 0.005 (0.05/10). CONTROL was significantly lower than all other conditions, 5-MIN (*d* = 0.69, *p* < 0.001); 10-MIN (*d* = 0.50, *p* < 0.001); 15-MIN (*d* = 0.72, *p* < 0.001); 20-MIN (*d* = 0.67, *p* < 0.001). No differences were found between the SPB conditions.

Regarding the main effect of time, three follow-up post-hoc *t*-tests were conducted, adjusting alpha level with Bonferroni correction to 0.017 (0.05/3). DURING was significantly higher than PRE (*d* = 1.455, *p* < 0.001) and POST (*d* = 1.355, *p* < 0.001), while there was a tendency for POST to be higher than PRE (*d* = 0.364, *p* = 0.021).

Regarding the interaction effect condition * time, based on our hypotheses, we focused on the following comparisons: All conditions compared at each of the three time points (i.e., PRE, DURING, and POST), resulting in 30 comparisons, adjusting alpha level with Bonferroni correction to 0.001 (0.05/30). At PRE, the five conditions did not differ from one another. At DURING, all SPB conditions were higher than CONTROL (5-MIN: *d* = −1.218, *p* < 0.001; 10-MIN: *d* = −1.137, *p* < 0.001; 15-MIN: *d* = −1.255, *p* < 0.001; 20-MIN: *d* = −1.173, *p* < 0.001), but did not differ among themselves (all comparisons with *p* > 0.001). At POST, the SPB conditions did not differ from the control, and did not differ among themselves.

### 3.2. Manipulation Check: Respiratory Frequency (RF)

For RF, we found a significant main effect of condition *F*(2.141, 124.207) = 941.021, *p* < 0.001, partial *η*^2^ = 0.18; a significant main effect of time *F*(1.615, 93.693) = 896.034, *p* < 0.001, partial *η*^2^ = 0.60; and a significant time * condition interaction effect *F*(3.490, 202.407) = 486.618, *p* < 0.001, partial *η*^2^ = 0.15.

Regarding the main effect of condition, 10 follow-up post-hoc t-tests were conducted, adjusting alpha level with Bonferroni correction to 0.005 (0.05/10). RF was higher in CONTROL than in all other conditions, 5-MIN (*d* = 4.60, *p* < 0.001); 10-MIN (*d* = 4.622, *p* < 0.001); 15-MIN (*d* = 4.548, *p* < 0.001); 20-MIN (*d* = 5.048, *p* < 0.001). RF was significantly higher in the 5-MIN condition than in the 10-MIN condition (*d* = 0.571, *p* < 0.001); 15-MIN condition (*d* = 0.87, *p* < 0.001); and 20-MIN condition (*d* = 1.543, *p* < 0.001). RF was significantly higher in the 10-MIN condition than in the 20-MIN condition (*d* = 1.162, *p* < 0.001), but not different from the 15-MIN condition (*d* = 0.428, *p* = 0.017). Finally, RF was significantly higher in the 15-MIN than in the 20-MIN condition (*d* = 0.970, *p* < 0.001).

Regarding the main effect of time, three follow-up post-hoc t-tests were conducted, adjusting alpha level with Bonferroni correction to 0.017 (0.05/3). RF was significantly lower in DURING than in PRE (*d* = 5.317, *p* < 0.001) and POST (*d* = 3.100, *p* < 0.001). Finally, RF was significantly lower in POST than in PRE (*d* = 3.100, *p* = 0.021).

Regarding the interaction effect condition * time, based on our hypotheses, we focused on the following comparisons: All conditions compared at PRE, DURING, and POST, resulting in 30 comparisons, with the alpha level adjusted with Bonferroni correction to 0.001 (0.05/30). For PRE, RF did not differ among the five conditions. For DURING, RF was lower in all SPB conditions than in CONTROL (5-MIN, *d* = 5.456, *p* < 0.001; 10-MIN, *d* = 5.342, *p* < 0.001; 15-MIN, *d* = 5.160, *p* < 0.001; 20-MIN, *d* = 5.190, *p* < 0.001), but did not differ among the SPB conditions (all comparisons with *p* > 0.001). At POST, RF was lower in all SPB conditions than in CONTROL, 5-MIN (*d* = 1.352, *p* < 0.001); 10-MIN, *d* = 1.733, *p* < 0.001; 15-MIN, *d* = 2.102, *p* < 0.001; 20-MIN, *d* = 2.836, *p* < 0.001. Moreover, RF after 5-MIN was higher than RF after 10-MIN (*d* = 0.897, *p* < 0.001); 15-MIN (*d* = 1.528, *p* < 0.001); and 20-MIN (*d* = 1.924, *p* < 0.001). RF after 10-MIN was higher than RF after 15-MIN (*d* = 0.943, *p* < 0.001) and 20-MIN (*d* = 1.604, *p* < 0.001). Finally, RF after 15-MIN was higher than 20-MIN (*d* = 1.069, *p* < 0.001).

## 4. Discussion

The aim of this study was to investigate the dose-response relationship of SPB without biofeedback on CVA. Our first hypothesis was validated, as CVA increased in all SPB conditions in comparison to the CONTROL condition. We did not have a specific hypotheses concerning the dose-response relationship, and findings revealed that no difference was found during SPB while increasing its duration, nor immediately after stopping SPB. Additionally, our manipulation check revealed that the duration of SPB had an influence on the spontaneous respiratory frequency immediately after SPB, in that the longer the SPB session lasted, the lower the spontaneous respiratory frequency was after performing SPB.

The finding that CVA increases during SPB and decreases afterwards corresponds to the previous research [23,31,32,45]. This increase in CVA, reflects the suggested impact of SPB on the vagus nerve [3,25]. It should be noted, that the different aforementioned mechanisms could all be related to some extent to the central autonomic network [19] and to the neurovisceral integration model [17,18], which may provide an explanation for the relationship between SPB and CVA increases. The finding that CVA returns to baseline levels when SPB stops, depicts a type of a “turn-on/turn-off” mechanism associated first with the stimulation and then with the cessation of the vagus nerve stimulation. Future research is needed in order to investigate whether the effects of SPB on other psychophysiological parameters remain after the cessation of a SPB practice.

Overall, a dose-response relationship between SPB and CVA has neither been found during SPB (i.e., the last 5-min of SPB in each condition) nor immediately after SPB. Consequently, the findings indicate a no dose-response effect of SPB on CVA within one SPB session. Our study was, to the best of our knowledge, the first to investigate the dose-response relationship effect of SPB on CVA. The meta-analysis of Lehrer et al. [48] tested the dose-response effect of SPB with biofeedback in long-term interventions (i.e., several sessions) on a range of psychophysiological outcomes (CVA was not part of them), through their moderator analysis, and results showed that effect sizes were not moderated by the length of treatment and home practice. Regarding other psychological techniques, such as mindfulness, a dose-response meta-regression [49] showed that there was no evidence of larger doses to be more helpful than smaller doses in predicting psychological outcomes. However, other moderators such as greater face-to-face contact, program intensity, and actual program use were found to be significant moderators. The role of these possible moderators should also be investigated in future studies.

Our manipulation checks on RF revealed that a dose-response was observed regarding this parameter, namely the longer the SPB session, the lower the RF immediately after SPB. This finding is of particular interest, given the fact that it may contribute to some lasting effects of SPB, despite the cessation of vagus nerve stimulation. This result is in line with previous findings, as decreases in spontaneous RF were observed in longer SPB training involving several sessions [50,51]. A lower spontaneous breathing rate is also related to a lower sympathetic sensitivity, potentially providing the support for more adaptive responses during stressful events [52]. Overall, given the fact that stress states are related to higher respiratory rates [53,54,55,56], and that slower respiratory frequency is related to relaxation outcomes [1], this may be an interesting direction for future research, as monitoring respiratory rates may enhance the predictive accuracy of a large range of positive and negative (health-related) outcomes [57].

Our study had some limitations. First, we always investigated the same breathing frequency (6 cpm) and the same inhalation/exhalation ratio (4 s/6 s). Previous research showed that the manipulation of these two parameters may influence CVA [5,29]. Future research should systematically investigate whether the manipulation of breathing parameters leads to different results, by combining different conditions for each parameter with one another. Second, the influence of a post-inhalation and post-exhalation respiratory pause should also be investigated, given the fact that the previous research [58] mentioned that a post-exhalation respiratory pause of 4 s may increase CVA. However, caution should be taken when interpreting this finding, given that the HRV parameter on which the authors based their conclusion, i.e., high-frequency HRV, does not reflect CVA during SPB [59]. Additionally, other research also showed that a brief (0.4 s) post-inhalation and post-inhalation pause had no influence on CVA as indexed by RMSSD [5]. Third, the question regarding whether practicing SPB at the resonance frequency (i.e., the respiratory frequency suggested to provoke the strongest stimulation of the baroreflex) may trigger different effects on CVA should also be addressed [27]. However, to date no clear benefits could be attributed to performing SPB at the resonance frequency [48] rather than at the standard rate of 6 cpm. Finally, our finding related to RF as a manipulation check should be considered cautiously, given the fact that the algorithm used to compute RF is a post-hoc assessment based on the ECG signal. Even if this method has been deemed reliable [37], future research should consider more direct measurements with respiratory belts, and potentially consider measuring additionally other respiratory parameters, such as respiratory depth or the partial pressure of end-tidal carbon dioxide, which might help in detecting hyperventilation.

## 5. Conclusions

To conclude, our findings replicate the previous finding that CVA increases during SPB without biofeedback, and decreases after terminating SPB. Nonetheless, the effects of SPB on other physiological variables may persist longer, given that the spontaneous respiratory frequency measured at rest after SPB was found to be lower than before SPB, the decrease being larger with a longer SPB duration. Consequently, the finding that the SPB duration did not impact CVA does not exclude the impact of duration on other psychophysiological parameters, which should be investigated in future research. As an example, positive psychological outcomes, such as improved executive functions, were observed immediately after implementing SPB without biofeedback in athletes in resting conditions [31,32] and after physical exertion [45], while emotion regulation was improved in clinical populations [38]. Taken together, our findings illustrated that even small acute doses of SPB without biofeedback can be beneficial in activating the “vagal brake”, and potentially in triggering positive physiological outcomes. These findings promote SPB as an effective strategy to decrease respiratory frequency, even during short intervals, while long-term interventions may provide chronic increases in CVA [24].

## Figures and Tables

**Figure 1 ijerph-18-12478-f001:**
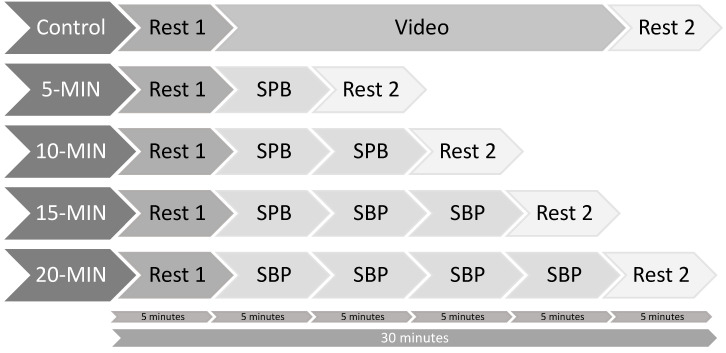
Experimental protocol. Note: SPB: Slow-paced breathing = 5 min; Rest = 5 min.

**Table 1 ijerph-18-12478-t001:** Descriptive statistics.

		PRE		DURING		POST	
		M	SD	M	SD	M	SD
RMSSD	Control	60.58	38.55	58.56	35.87	57.52	36.07
	5-MIN	56.82	32.10	82.68	35.37	60.64	30.75
	10-MIN	55.61	34.02	83.35	37.52	60.34	35.47
	15-MIN	58.15	35.04	86.13	37.14	63.24	32.78
	20-MIN	57.91	34.79	83.95	37.64	62.33	34.52
	Control (Log)	1.71	0.25	1.70	0.24	1.68	0.27
	5-MIN (Log)	1.69	0.24	1.88	0.19	1.73	0.22
	10-MIN (Log)	1.67	0.25	1.88	0.19	1.71	0.25
	15-MIN (Log)	1.69	0.26	1.90	0.19	1.74	0.23
	20-MIN (Log)	1.69	0.25	1.88	0.19	1.73	0.24
Respiratory Frequency	Control	17.09	2.61	16.90	2.73	17.02	2.53
	5-MIN	16.78	3.03	6.58	0.28	14.18	3.02
	10-MIN	16.71	2.93	6.57	0.27	13.16	2.70
	15-MIN	17.05	2.67	6.53	0.30	12.16	2.57
	20-MIN	16.91	2.78	6.50	0.28	11.14	2.30
	Control (Log)	1.23	0.07	1.22	0.08	1.23	0.07
	5-MIN (Log)	1.22	0.08	0.82	0.02	1.14	0.10
	10-MIN (Log)	1.22	0.08	0.82	0.02	1.11	0.09
	15-MIN (Log)	1.23	0.07	0.81	0.02	1.08	0.09
	20-MIN (Log)	1.22	0.08	0.81	0.02	1.04	0.09
Heart Rate	Control	66.18	10.61	67.62	10.32	67.39	10.36
	5-MIN	67.70	10.44	69.13	7.90	67.88	9.21
	10-MIN	66.18	10.33	67.76	7.58	67.70	10.33
	15-MIN	65.41	10.98	67.78	8.37	68.02	10.05
	20-MIN	66.79	10.46	67.61	8.65	67.57	10.40
	Control (Log)	1.82	0.07	1.83	0.07	1.82	0.07
	5-MIN (Log)	1.83	0.07	1.84	0.05	1.83	0.06
	10-MIN (Log)	1.82	0.07	1.83	0.05	1.83	0.07
	15-MIN (Log)	1.81	0.07	1.83	0.06	1.83	0.06
	20-MIN (Log)	1.82	0.07	1.83	0.06	1.82	0.07

Note: RMSSD: Root mean square of successive differences.

**Table 2 ijerph-18-12478-t002:** Mean comparisons regarding time—pre, during, and post.

		Difference PRE/DURING		Difference PRE/POST		Difference DURING/POST	
		Cohen’s *d*	*p*	Cohen’s *d*	*p*	Cohen’s *d*	*p*
RMSSD	Control	0.167	0.204	0.292	0.029	0.203	0.124
	5-MIN	**−1.305**	**<0.001**	−0.363	0.007	**1.349**	**<0.001**
	10-MIN	**−1.445**	**<0.001**	−0.284	0.033	**1.262**	**<0.001**
	15-MIN	**−1.219**	**<0.001**	−0.411	0.003	**1.062**	**<0.001**
	20-MIN	**−1.350**	**<0.001**	−0.383	0.005	**1.104**	**<0.001**
Respiratory frequency	Control	0.185	0.162	0.024	0.857	−0.148	0.260
	5-MIN	**4.696**	**<0.001**	**1.308**	**<0.001**	**−3.317**	**<0.001**
	10-MIN	**4.710**	**<0.001**	**1.745**	**<0.001**	**−3.147**	**<0.001**
	15-MIN	**5.459**	**<0.001**	**2.014**	**<0.001**	**−2.774**	**<0.001**
	20-MIN	**4.987**	**<0.001**	**2.963**	**<0.001**	**−2.468**	**<0.001**
Heart Rate	Control	−0.199	0.131	−0.179	0.174	0.025	0.846
	5-MIN	−0.245	0.065	−0.049	0.711	0.196	0.138
	10-MIN	−0.217	0.101	−0.151	0.250	0.062	0.634
	15-MIN	−0.269	0.043	−0.389	0.004	−0.003	0.984
	20-MIN	−0.126	0.338	−0.098	0.452	0.037	0.776

Note: Significant differences are indicated in bold; RMSSD: Root mean square of successive differences.

**Table 3 ijerph-18-12478-t003:** Mean comparisons between conditions—control, 5-MIN, 10-MIN, 15-MIN, 20-MIN.

		PRE		DURING		POST	
		Cohen’s *d*	*p*	Cohen’s *d*	*p*	Cohen’s *d*	*p*
RMSSD	Control—5-MIN	0.152	0.249	**−1.218**	**<0.001**	−0.370	0.006
	Control—10-MIN	0.248	0.062	**−1.137**	**<0.001**	−0.194	0.142
	Control—15-MIN	0.149	0.256	**−1.255**	**<0.001**	−0.402	0.003
	Control—20-MIN	0.127	0.335	**−1.173**	**<0.001**	−0.376	0.005
	5-MIN—10-MIN	0.113	0.389	−0.014	0.915	0.179	0.173
	5-MIN—15-MIN	0.001	0.992	−0.221	0.095	−0.143	0.278
	5-MIN—20-MIN	−0.030	0.818	−0.058	0.655	−0.037	0.780
	10-MIN—15-MIN	−0.141	0.282	−0.211	0.110	−0.306	0.022
	10-MIN—20-MIN	−0.175	0.185	−0.035	0.790	−0.247	0.063
	15-MIN—20-MIN	−0.037	0.778	0.202	0.125	0.108	0.411
Respiratory frequency	Control—5-MIN	0.217	0.102	**5.456**	**<0.001**	**1.352**	**<0.001**
	Control—10-MIN	0.294	0.028	**5.342**	**<0.001**	**1.733**	**<0.001**
	Control—15-MIN	0.032	0.804	**5.160**	**<0.001**	**2.102**	**<0.001**
	Control—20-MIN	0.135	0.304	**5.190**	**<0.001**	**2.836**	**<0.001**
	5-MIN—10-MIN	0.046	0.723	0.047	0.722	**0.897**	**<0.001**
	5-MIN—15-MIN	−0.188	0.155	0.170	0.196	**1.528**	**<0.001**
	5-MIN—20-MIN	−0.126	0.337	0.274	0.040	**1.924**	**<0.001**
	10-MIN—15-MIN	−0.253	0.057	0.152	0.247	**0.943**	**<0.001**
	10-MIN—20-MIN	−0.204	0.122	0.298	0.026	**1.604**	**<0.001**
	15-MIN—20-MIN	0.111	0.398	0.114	0.384	**1.069**	**<0.001**
Heart Rate	Control—5-MIN	−0.195	0.139	−0.242	0.068	−0.067	0.610
	Control—10-MIN	−0.009	0.942	−0.057	0.666	−0.009	0.945
	Control—15-MIN	0.103	0.431	−0.041	0.755	−0.070	0.594
	Control—20-MIN	−0.084	0.522	−0.029	0.827	−0.008	0.951
	5-MIN—10-MIN	0.163	0.216	0.239	0.071	0.058	0.657
	5-MIN—15-MIN	0.239	0.071	0.189	0.153	−0.017	0.897
	5-MIN—20-MIN	0.102	0.436	0.267	0.045	0.047	0.717
	10-MIN—15-MIN	0.113	0.389	0.009	0.948	−0.054	0.679
	10-MIN—20-MIN	−0.085	0.516	0.047	0.719	<0.001	1.000
	15-MIN—20-MIN	−0.210	0.112	0.027	0.839	0.063	0.632

Note: Significant differences are indicated in bold; RMSSD: Root mean square of successive differences.

## Data Availability

Data can be shared by the corresponding author upon reasonable request.

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
