# Peer review of "Single Slow-Paced Breathing Session at Six Cycles per Minute: Investigation of Dose-Response Relationship on Cardiac Vagal Activity"

_ijerph, 2021, doi:10.3390/ijerph182312478_

Round 1

Reviewer 1 Report

The authors  conducted a well-done experiment on 59 participants in which they demonstrated that the duration of slow-paced breathing sessions did not have an effect on cardiac vagal activity, i.e., there was no dose-response effect. The article is very well written and requires no changes or corrections. It was a pleasure to read.

Suggested addition to the Discussion

A possible weakness of the study is that nasal vs oral breathing during the SPB exercise was not considered. Although the “Participants were instructed to inhale continuously through the nose while the ball was moving upwards, and exhale continuously with pursed lips when the ball was moving downwards” it is not clear (or it is unknown) whether they were actually breathing nasally, and if so, to what extent.  There is a growing body of evidence that various aspects of brain function may be optimized with nasal breathing, see for example, Zelano et al (2016). A consideration of this issue would enrich the discussion, including how it might be operationally addressed in future research on SPB.

Zelano, C.; Jiang, H.; Zhou, G.; Arora, N.; Schuele, S.; Rosenow, J.; Gottfried, J.A. Nasal Respiration Entrains Human Limbic Oscillations and Modulates Cognitive Function. J. Neurosci. 2016, 36, 12448–12467.

Reviewer 2 Report

I would like to discuss some issues that may need to be addressed in the manuscript. They are listed below not necessarily in order of importance.

1. Procedure section 
- How were patients randomized to 5 sessions?
Please explain the procedure in detail.

2. Table 1.
- Please indicate the significant difference between pre, during, and post(time)in the table.

- Please indicate the significant difference between control, 5-min, 10-min, 15-min, 20-min(condition) in the table.
